# Development of an Autonomous Underwater Helicopter with High Maneuverability

**Zhikun Wang, Xun Liu, Haocai Huang *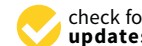 and Ying Chen**

Ocean College, Zhejiang University, Zhoushan 316021, China; 11834011@zju.edu.cn (Z.W.);
21734116@zju.edu.cn (X.L.); ychen@zju.edu.cn (Y.C.)
* Correspondence: hchuang@zju.edu.cn; Tel.: +86-58-0209-2203

**Abstract:** Autonomous Underwater Vehicles (AUVs) are the mainstream equipment for underwater scientific research and engineering. However, it remains a great challenge for AUVs to carry out near-seabed operations because of their poor maneuverability. In this paper, a new design for a high-maneuverability disc-shaped AUV is proposed, namely, the Autonomous Underwater Helicopter (AUH). We designed the AUH's propulsion system through dynamic analysis based on the unique disc shape. The experimental prototype was built by mechatronics technology, after which several motion experiments were carried out to demonstrate the high maneuverability. We find that the prototype has high maneuverability: it can cruise at 0.8 m/s (about 1.5 knots), at least; its turning radius is zero and its turning speed is at least 20 deg/s; and the motion of specific curves in a small range was completed. It is demonstrated that over-actuation is not necessary for the high-maneuverability AUH because of its unique disc shape. A propulsion system consisting of four propellers and a buoyancy adjustment system is used for the highly maneuverable AUH. In addition, the AUH may be a solution for near-seafloor operations.

**Keywords:** autonomous underwater vehicle; disc-shape; dynamic analysis; high maneuverability; propulsion system

---

## 1. Introduction

In deep-sea exploration, Autonomous Underwater Vehicles (AUVs) play an important role by virtue of their unique characteristics, which include their autonomous, long-time and long-range operation capabilities [1]. The research on them is booming all over the world, and they have caused a revolution in the field of ocean research [2]. On one hand, the demand for AUVs comes from the need for large-scale underwater observation and resource exploration in traditional scientific research. On the other hand, based on the application and development of seabed infrastructure equipment and various types of underwater machinery, the maintenance of seabed constructions also puts forward new requirements for AUVs [3]. These tasks require high maneuverability in small areas and stability while performing tasks. To satisfy various applications, including bionic submarines, various forms of AUVs have been proposed [2]. However, near-seabed operation remains a great challenge for AUVs.

Shape is very important for the maneuverability of AUVs. The requirements of traditional AUVs are long-range, high-speed and long-duration operation. Therefore, most research has been focused on one-axis streamlined underwater vehicles, such as torpedo-shaped submersibles of the water-droplet type or the long cylindrical type [4]. Although these types of AUV are suitable for long-distance navigation [5], most of them have poor maneuverability [6]. Obviously, it is difficult for them to adapt to new modes of work that combine large-scale cruising over several kilometers with small-scale flexible operation over a few meters.

High maneuverability in the horizontal plane and stability in the vertical direction are required in small-scale operations. The disc shape has one prominent advantage. It has the characteristics of good motion stability at low speeds [7]. Furthermore, disc-shaped AUVs have several unique advantages: they are highly maneuverable in harsh working environments; they can explore from all angles and directions within a narrow space; their utilization of internal space is more efficient; their completely symmetrical shape is not easily caught by weeds during navigation [8]; and the resistance caused by flow in all directions is low due to the centrally symmetrical shape [9].

The research on disc-shaped AUVs is still in the laboratory, and most mature disc-shaped AUVs are glider forms, which only use buoyancy and gravity for long-distance navigation, similar to the principle of a glider in the air. With CFD technology, a variety of more mature underwater gliders have been developed. The motion characteristics of classic underwater gliders have been studied to some extent using CFD technology [10], such as the CIDESI Underwater Glider [11]. The first disc-shaped AUV was Discuz of the Webb Corporation in the United States. It has been demonstrated to be capable of resisting the disturbance of advection and landing on the seabed for observation [12]. In addition, there are LUNA [13] and BOOMERANGE [14] of the Research Institute for Applied Mechanics, Kyushu University, and a disc-shaped underwater glider of Dalian Maritime University [15]. These disc-shaped AUVs utilize the isotropic properties of the disc shape in resisting plane flow and achieving excellent steering characteristics. However, due to the lack of active driving, they cannot carry out complex and flexible planar motion. For example, BOOMERANG is almost unable to move in the horizontal plane, and its steering speed is about 5 deg/s [16]. In addition, for the disc-shaped glider made by the Dalian Maritime University, it takes nearly $40 \times 40$ m space to make a 180° turn.

To be maneuverable in the horizontal plane, we consider the application of the disc shape to the common AUV with propulsion components and propose a new disc-shaped AUV, namely, the Autonomous Underwater Helicopter (AUH). It is so named as it can cruise in a small area flexibly, hover at a fixed point, land on the seabed and take off, similar to the capabilities of a conventional helicopter [3].

In Section 2, we introduce the working pattern of AUH in detail and elaborate on the differences between the AUH and other AUV. In Section 3, the specific dynamic analysis of the AUH is elaborated. Depending on the dynamic analysis, a propulsion system is proposed that is as simple as possible under the premise of maintaining high maneuverability. The propulsion system consists of four propellers and a buoyancy adjustment system. In Section 4, the electromechanical system of a prototype AUH is designed in detail to obtain higher maneuverability, consisting of a unique disc-shaped hull, a propulsion system, a control system, a positioning and navigation system, etc. In Section 5, we introduce the AUH's motion experiments and the results, and the high maneuverability of AUH is verified. Finally, concluding remarks are given in Section 6.

## 2. The Working Pattern of AUH

Figure 1 shows the main working pattern of the AUH. It works from seabed to seabed, completing missions such as communication, charging, equipment maintenance, etc. The AUH starts from the surface and dives to working depth using a buoyancy adjustment system. Under the guidance of the positioning system, it cruises from one Subsea Station to the other, and then lands on or hovers over it to complete the missions. Accurate motion above the base station requires the AUH to have good maneuverability in the plane, while stable hovering over the base station requires AUH stability in the vertical direction. The Subsea Station is a base station for the AUH when performing functions such as data interaction and power transmission. Because of its distinctive disc-shaped body of revolution, the AUH can steer flexibly. It also has low damping in horizontal motion, but high damping in vertical motion. These characteristics make AUH have unique advantages in precise landing and stable hovering operation. After completing the missions, the AUH can take off and go to the next base station by means of the positioning and navigation system. Finally, after all missions are done, the AUH can come up to the surface using the buoyancy adjustment system.

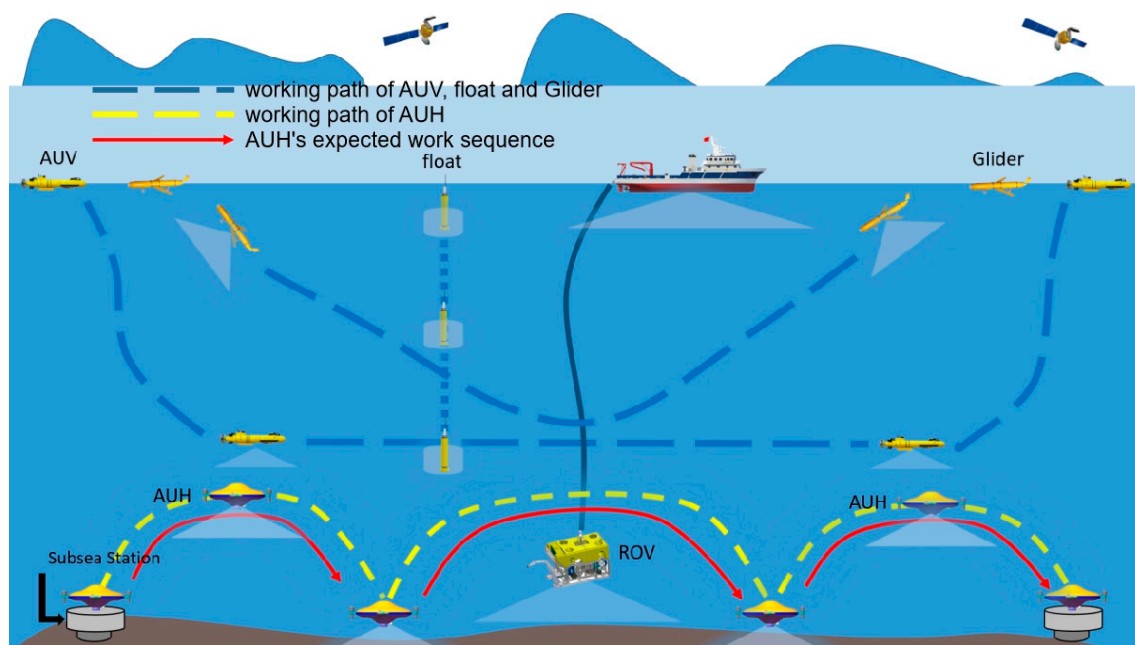

**Figure 1.** AUH's main working pattern and the differences between AUH and other underwater vehicles.

To meet the requirements of the main working pattern, the AUH must be maneuverable in the horizontal plane and stable in the vertical direction.

## 3. Dynamic Analysis of the AUH

Steady linear motion, fast turning, and small turning radius are important requirements for high maneuverability. Generally speaking, for underwater vehicles, a typical six-degree-of-freedom system with over-actuation is a prerequisite for achieving high maneuverability. For example, the hovering type AUV 'cyclops' has 8 thrusters in order to achieve high maneuverability [17]. However, too many propulsion components will compromise the good hydrodynamic characteristics of the disc shape and increase the loss of energy, as well. Therefore, based on the dynamic analysis in this section, we will attempt to achieve high maneuverability of the AUH with as few possible propulsion components as possible.

### 3.1. Dynamic Model of the AUH

To describe the motion conveniently, the coordinate system is established as shown in Figure 2. According to the momentum theorem, the linear motion description of AUH can easily be described as:

$$\mathbf{F} = \frac{d\mathbf{H}}{dt} = \frac{d(m\mathbf{V}_{GE})}{dt} \tag{1}$$

where $\mathbf{F}$ is the resultant external force vector. $\mathbf{H}$ is the momentum of AUH. $\mathbf{V}_{GE}$ is the velocity of AUH relative to fixed coordinate system E-xyz. Through the transformation between coordinate systems, the equation of motion in moving coordinate system o-pqr can be described as:

$$\mathbf{F} = m\frac{d(\mathbf{V} + \boldsymbol{\omega} \times \mathbf{R}_{Go})}{dt} = m\left[\frac{\partial \mathbf{V}}{\partial t} + \boldsymbol{\omega} \times \mathbf{V} + \frac{\partial \boldsymbol{\omega}}{\partial t} \times \mathbf{R}_{Go} + \boldsymbol{\omega} \times (\boldsymbol{\omega} \times \mathbf{R}_{Go})\right] \tag{2}$$

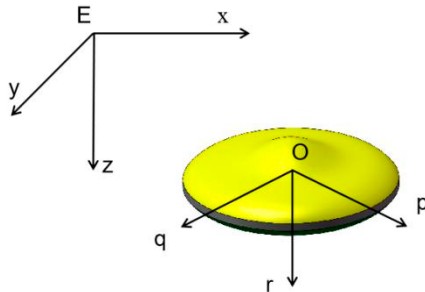

**Figure 2.** Fixed coordinate system and moving coordinate system.

According to Euler's second law, the AUH rotational motion is described as:

$$\mathbf{T} = \frac{d\mathbf{L}}{dt} = \mathbf{J}\frac{\partial \boldsymbol{\omega}}{\partial t} + \boldsymbol{\omega} \times (\mathbf{J}\boldsymbol{\omega}) + \mathbf{R}_{Go} \times m\frac{\partial \mathbf{V}_{Go}}{\partial t} + \mathbf{R}_{Go} \times (\boldsymbol{\omega} \times m\mathbf{V}_{Go}) \tag{3}$$

where $\mathbf{T}$ is the resultant external moment vector relative to the origin of the moving coordinate system. $\mathbf{L}$ is the Momentum moment of AUH relative to fixed coordinate system E-pqr. $\mathbf{R}_{Go}$ and $\mathbf{V}_{Go}$ are distance and velocity of AUH's center of gravity relative to o-pqr, respectively. $\mathbf{J}$ is the moment of inertia. In Equation (3), the first term represents the moment of inertia of rotating motion; the second term represents the moment of inertia of rotating unbalanced rotation of rotating axis; the latter is benign because the origin is not in the center of gravity; the third term is the moment of inertia of linear motion; and the fourth term is the moment of inertia of centrifugal motion.

The complete description of AUH motion can be obtained by Equations (2) and (3). Next, we design the propulsion system by force analysis as simply as possible.

### 3.2. Analysis of Motion in the Horizontal Plane

The main three special functions of AUH's working mode are short-range flexible motion, fixed-point hovering, and landing and take-off. This means that the AUH has a need for 6 DOF control.

The force analysis of horizontal motion is shown in Figure 3. The equation of linear motion is:

$$F_1 + F_2 - f_p = m\frac{du}{dt} \tag{4}$$

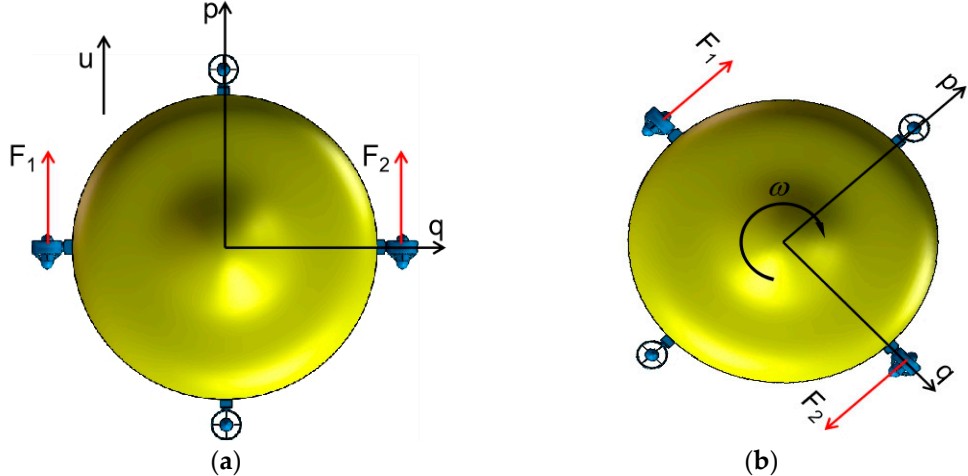

(**a**)             (**b**)

**Figure 3.** Force analysis of horizontal motion. (**a**) Linear motion; (**b**) rotation motion.

The equation of rotation motion is:

$$F_1 R + F_2 R - M_{fr} = J_r \frac{d\omega}{dt} \tag{5}$$

where $F_1$ and $F_2$ are the force of the horizontal thrusters. $f_p$ is the resistance of flow. $M_{fp}$ is the resistance moment of the z-axis. $u$ is the velocity of the p-axis. $\omega$ is the angular velocity about the r-axis. $R$ is the radius of AUH, which is effectively the moment arm for each thruster.

The resistance of AUH is mainly inertial fluid force and viscous fluid force. Since inertial fluid force can be regarded as additional mass, we mainly discuss viscous fluid force. Viscous fluid force can be divided into viscous resistance related to the velocity and contact area of the submarine and differential pressure resistance related to the shape of the submarine. Because the AUH presents a remarkable non-streamlined line, this may make the viscous resistance in translation direction negligible for differential pressure resistance [18]. We can get

$$f_p = C_{u|u|}|u| + C_u S u \approx C_{u|u|}|u| \tag{6}$$

where $C_{u|u|}$ is the quadratic differential pressure coefficient, $C_u$ is translational viscous drag coefficient, S is the surface area of AUH. In the rotary motion, due to the rotatory characteristics of the AUH, there is only viscous resistance and no differential pressure resistance. So

$$M_{fr} = \int_S C_\omega \omega r dr \tag{7}$$

where $C_u$ is the rotary viscous drag coefficient. Since $C_u$ and $C_\omega$ are only related to fluid properties and shell material, they are of the same order of magnitude. When $u$ is the same as $\omega r$, $M_{fr}$ is the same order of magnitude as $C_u S u R$. The AUH design translation speed is 0.8 m/s. Assuming that the AUH edge has a rotational linear velocity of 0.8 m/s, its angular velocity is about 90 deg/s, which is sufficient for high maneuverability. The thrust of the propeller selected according to the translational design speed is at least $f_p$, and the minimum torque $M_{fr}$ required to maintain the design rotational speed is much smaller than the torque $2f_p R$ that the propeller can provide. Therefore, $M_{fr}$ can be ignored in the analysis.

Higher maneuverability means that linear motion and rotary motion have faster responses. From Equations (4) and (5), it can be seen that higher $F_1$, $F_2$, $R$ and smaller $f_p$, $M_{fr}$, $m$, and $J_r$ mean higher response speed. It can be seen from the above that $M_{fr}$ is negligible. So the response speed of rotation is directly correlated with $J_r$. Because of the special shape of AUH, $J_r$ can be quite small when the mass distribution is concentrated near the center as much as possible. Therefore, AUH has good horizontal plane rotation ability. In addition, AUH can achieve zero radius rotation by providing a set of symmetrical opposite driving forces.

The response of plane motion is similar to that of rotation motion, but the difference is that $f_x$ is not zero and the situation is more complex. The analysis is as follows;

When AUH advances under the action of a horizontal propeller, the force acting on it is shown in Figure 4. Considering stability, AUH's center of gravity is lower than the center of buoyancy. In addition, the special requirement of positioning and navigation system leads to the change of body shape. Both of them will cause horizontal thrust $F_h$ and fluid resistance $f_p$ to produce moments $M_{Fh}$ and $M_{fp}$ that will cause the AUH to pitch. After pitching, due to the distance between the center of gravity and the center of buoyancy, the restoring moment will be generated, and the $M_{fp}$ will change with the appearance of the angle of attack $\alpha$. The motion description of the pitch axis is shown in Equation (8).

$$-GR_{BG} \sin\theta + M_{Fp} + M_{fp} - M_{fq} = J_q \frac{d^2\theta}{dt^2} \tag{8}$$

where $G$ is gravity force. $R_{BG}$ is the distance of the center of gravity and the center of buoyancy. $\theta$ is the pitch angle. $J_q$ is the AUH's rotating inertia of pitch axis. $M_{fq}$ is the hydrodynamic resistance moment caused by the pitch. It can be approximately calculated by Equation (9) [19]:

$$M_{fq} = \int_0^R \rho C_d \frac{d\theta}{dt} \sqrt{R - x^2} x dx = \frac{1}{3} \rho C_d R^{\frac{3}{2}} \frac{d\theta}{dt} \tag{9}$$

where $\rho$ is the water density. $C_d$ is the resistance coefficient. $R$ is the radius of AUH. Bringing Equation (9) into Equation (8):

$$\begin{cases} A \sin\theta + D + E\dot{\theta} = H\ddot{\theta} \\ A = -GR_{BG} \\ D = M_{Fp} + M_{fp} \\ E = -\frac{1}{3}\rho C_d R^{\frac{3}{2}} \\ H = J_q \end{cases} \tag{10}$$

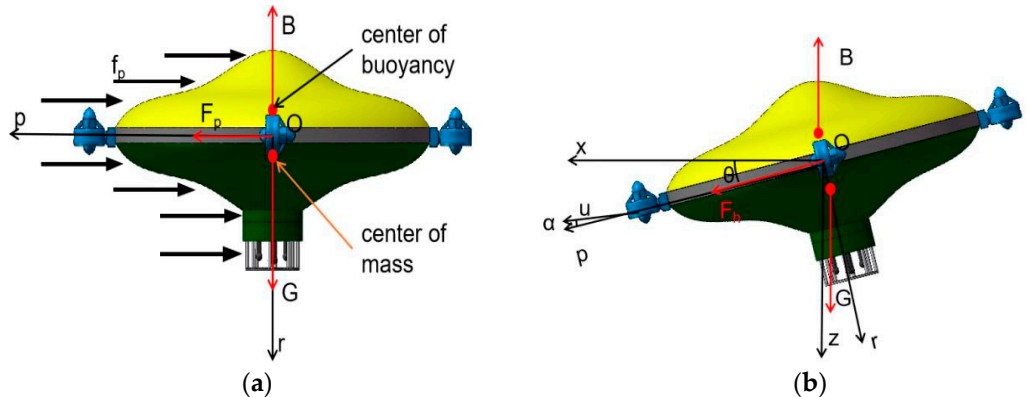

**Figure 4.** Force analysis of horizontal linear motion. (**a**) Initial state; (**b**) motion state.

To simplify the calculation, only consider the case where $\theta$ is small, and simplify $\sin\theta$ to $\theta$:

$$H\ddot{\theta} - E\dot{\theta} - D = A\theta \tag{11}$$

$$\theta = \frac{D}{2AK}\left[e^{t(E-K)/2H}(E+K) - e^{t(E+K)/2H}(E-K)\right] - \frac{D}{A}$$
$$K = \sqrt{E^2 + 4AH} \tag{12}$$

It can be seen from Equation (12) that since $E < 0$, when $K > 0$, the pitch motion is overdamped, and when $K < 0$, the pitch motion is underdamped. In addition a stable pitch angle will eventually be maintained. Due to the existence of inertia, when there is a pitch angle, there is generally an angle of attack $\alpha$. Unlike traditional torpedo submarines, pitch and attack angles are harmful to AUH. $f_p$ is directly related to $\alpha$. The flat disc shape causes the AUH to have greater fluid resistance in the presence of an angle of attack, and the non-zero pitch angle will prevent the AUH from moving in the plane. Therefore, a set of active recovery moments is necessary for high maneuverability. At the same time, for constant-depth stable hovering, the driving force in the z-direction is essential. A set of vertical propellers can be added in addition, which can meet the requirements of fixed depth hovering and pitch recovery moment in the z-direction.

### 3.3. Analysis of Motion in the Vertical Direction

The landing and take-off functions have different requirements compared to forward and lateral propulsion. Stable landing process requires AUH to be subjected to stable downward force, but the design of underwater vehicles generally requires a slight positive buoyancy. It is undesirable that the power consumption is greatly increased by having to use the propellers to maintain contact

with the seabed when landing for a long time under the slight positive buoyancy, or conversely, by operating under the condition of the negative buoyancy. Therefore, in addition to the two groups of propellers, a buoyancy adjustment system is added to adjust the buoyancy to complete the stable landing and take-off.

In the case of the driving forces mentioned above, four DOFs are limited, and two DOFs are still free. The following analysis shows that the remaining two degrees of freedom needn't be limited. The two DOFs are the y-direction translation and the *x*-axis rotation. Because the y-direction translation can be limited, as described above, primarily the *x*-axis rotation is discussed.

*X*-axis rotation, i.e., rolling, is not a motion function that needs to be actively controlled, but an interference that needs to be suppressed in the motion of the AUH. There are two reasons why rolling does not need active force: (1) The source of rolling interference is less, and the intensity is small. (2) The damping of the rolling motion is great. Because the AUH mainly works near the seabed, its interference sources are mainly ocean current and internal waves. The deep-sea current is relatively stable. AUH's spindle-like structure in the horizontal plane causes it to be less resistant, therefore making it difficult for rolling moment to be produced in the current. The general scale of internal wave is larger, and the occurrence region is only in the stable density decomposition layer; therefore, it is difficult for a torque effect to be produced on the AUH, with its relatively small size. Therefore, most of the rolling interference is turbulence or collision. When AUH is subjected to the interference moment $M_t$, the swing motion is performed as shown in Figure 5. Vibration can be described by Equation (13).

$$- GR_{BG} \sin \psi - M_{fh} = J \frac{d^2 \psi}{dt^2} \tag{13}$$

where $\psi$ is the angle of roll. $M_{fh} = \rho C_d R^{1.5} (d\psi/dt)^2$ is a moment generated by fluid resistance $f_h$. Hence, Equation (9) can be adjusted to (14).

$$- GR_{BG} \sin \psi - \rho C_d R^{1.5} \frac{d\psi}{dt} = J \frac{d^2 \psi}{dt^2} \tag{14}$$

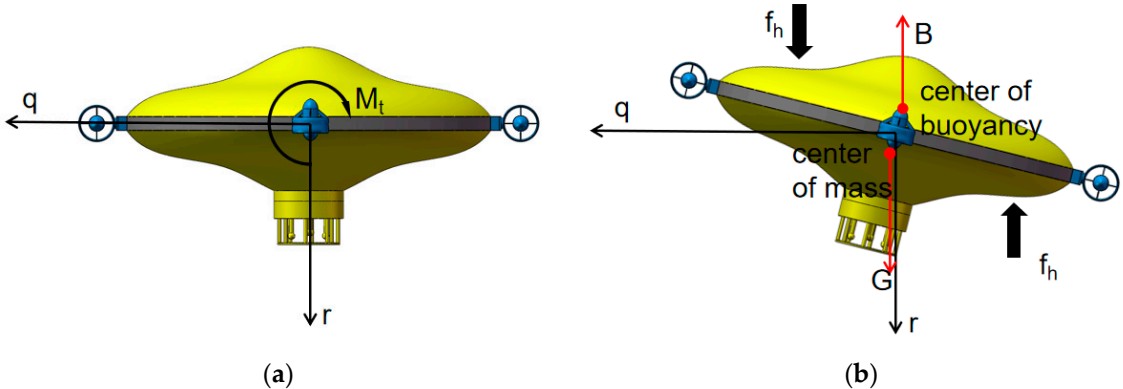

(**a**)　　　　　　　　　　　　　　　　　　　　(**b**)

**Figure 5.** *X*-axis disturbing vibration. (**a**) Initial state; (**b**) motion state.

Solving Equation (13) with reference to Equations (8)–(12) gives Equation (15):

$$\begin{cases} \psi = \frac{M_t J}{EK} \left( e^{t(E+K)/(2J)} - e^{t(E-K)/(2J)} \right) \\ A = -GR_{BG} \\ E = -\rho C_d R^{1.5} \\ K = \sqrt{E^2 + 4AH} \end{cases} \tag{15}$$

From Equation (15), similar to Equation (13), since $E < 0$, when $K > 0$, the roll motion is overdamped, and when $K < 0$, the pitch motion is underdamped. In addition, a zero roll angle will eventually be

maintained. As shown in Figure 5, because of the special structure of disc shape, the $M_{fh}$ is greater when the body is rolling, which means the damping term becomes greater. Therefore, the vibration decays rapidly and returns to a steady horizontal state without active force.

As mentioned above, using four thrusters in combination with the unique disc shape gives the AUH the characteristics of high maneuverability in the horizontal plane and a certain stability in the vertical direction. Based on the analysis, it is found that if the attack angle of horizontal navigation and the moment of inertia can be further decreased, the response of horizontal linear motion and rotation motion will be faster, which means having higher maneuverability.

## 4. Mechatronics Design of AUH Prototype

According to the dynamic analysis and the design of the propulsion system in Section 2, in order to realize horizontal maneuvering, vertical take-off and landing, we built a prototype of AUH. It is an electromechanical system composed of a control system, driving module, positioning and navigation system, and the unique disc-shaped hull.

### 4.1. General Layout

Considering the high maneuverability of the AUH disc shape and the zero-radius rotation characteristic of low damping, the symmetrical distribution of mass in the plane can achieve stability, and the arrangement of the large mass near the center is also beneficial to reducing the moment of inertia. This can further improve the maneuverability of AUH.

The AUH layout concept diagram and 3D model diagram are shown in Figure 6. It mainly includes two symmetrical cylindrical cabins, a battery cabin and a control system cabin, four propellers, iUSBL equipment for positioning and navigation, buoyancy material, a skeleton for connecting and supporting, and a disc-shaped hull. Each cabin passed the 100 m pressure seal test.

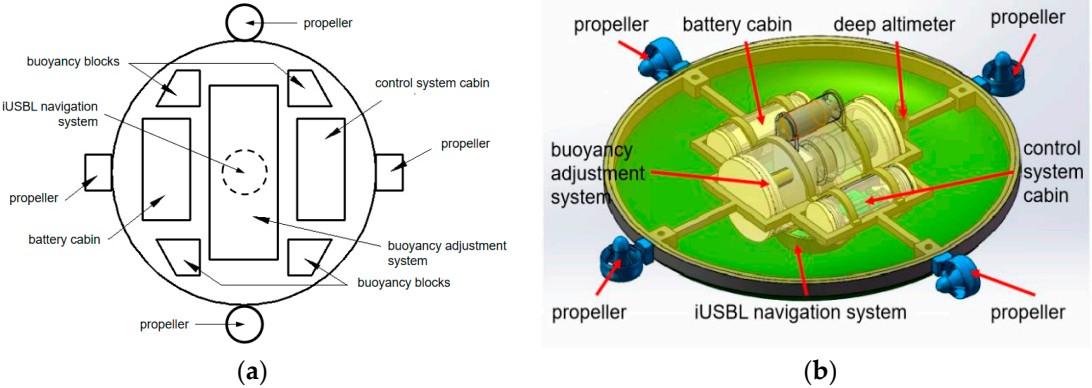

**Figure 6.** General layout of the prototype: (**a**) Top view layout concept diagram; (**b**) 3D model diagram.

The prototype is 1 m in diameter, 45 cm high and weighs 42 kg. The design speed is 0.8 m/s and the design depth is 100 m. Since the experimental prototype produced this time is only a prototype for verifying the working principle of the circular dish AUV, considering the test and iteration cost, the design depth is selected as 100 m, so that the design cost would remain low and the experimental environment would easier to find. In addition, the design speed is the estimated average flow rate in the working environment, considering that it should have a certain resistance to flow in order to achieve hovering.

### 4.2. Disc-Shaped Hull

The disc-shaped hull is one of the key designs of the Small-Autonomous Underwater Helicopter (S-AUH). To achieve the goal of high maneuverability in the horizontal plane and good stability in the vertical direction, the disc-shaped hull must be designed as a body of revolution, which is the reason

the hull is defined by a curve. After considering various curves, Non-Uniform Rational B-Splines (NURBS) were chosen because of the continuity of their curvature and their simple definitions, which makes the design and adjustment convenient. The spline is defined by 7 control points shown in Figure 7a. $L$ is the main size of the disc, the value is 1000 mm. $H$ is the direct parameter of the shape height. $h_1$ and $l_1$ is the parameter of the edge point. $h_2$ and $l_2$ is the parameter of the transition point. $h_3$ and $l_3$ is the parameter of the optimized point. According to the conclusions of [19], when $H = 200$ mm, $h_1 = 50$ mm, $h_2 = 80$ mm, $h_3 = 2(H - h_1)/3 + h_1 = 150$ mm, $l_1 = L/8 = 125$ mm, $l_2 = L/4 = 250$ mm, $l_3 = 3L/8 = 375$ mm, the movement in the horizontal plane has low resistance. In addition, in the vertical direction, it has good stability.

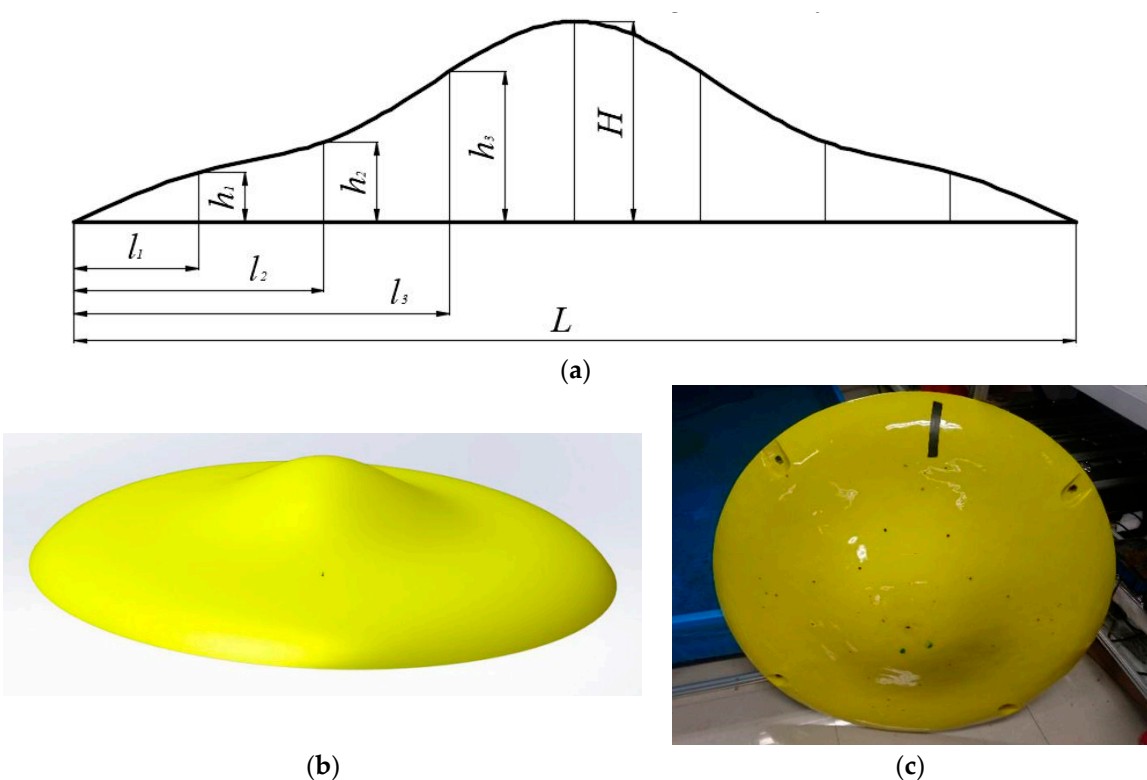

(a)

(b)                                                             (c)

**Figure 7.** (**a**) Parameters of the disc-shaped hull; (**b**) 3D model of the hull; (**c**) photo of the disc-shaped hull.

### 4.3. Propulsion Module

The propulsion module is responsible for providing propulsion force. As described in Section 2, four propellers and a set of buoyancy adjustment system are used to drive the AUH.

According to the results of hydrodynamic simulation of AUH in [18,20], the T200 propeller with performance as shown in Table 1 is selected. Its output thrust range meets the requirements for use because, by simulation analysis [18], it is considered that the thrust required for the design speed of 1 knot is about 4 N. In addition, the propeller we chose is light in weight, small in size and simple in mechanical connection.

**Table 1.** T200 propeller characteristics.

|  | 12 V | 16 V |
| --- | :---: | :---: |
| Max thrust (Forward) | 34.8 N | 50.0 N |
| Max thrust (Backward) | 29.4 N | 40.2 N |
| Min thrust | 0.098 N | |
| speed | 300–3800 $rev$/min | |
| Max current | 25 Amps | |
| Max power | 350 Watt | |

The buoyancy of the vehicle is adjusted by a piston-cylinder and oil bag system (Figure 8). The piston is driven by a DC motor, reduction gears, and a ball screw. Then the piston presses the hydraulic oil from the piston cylinder into the oil bag. Because the piston-cylinder and ball screw are located in the compressive cabin, the volume change of the oil bag leads to buoyancy variation.

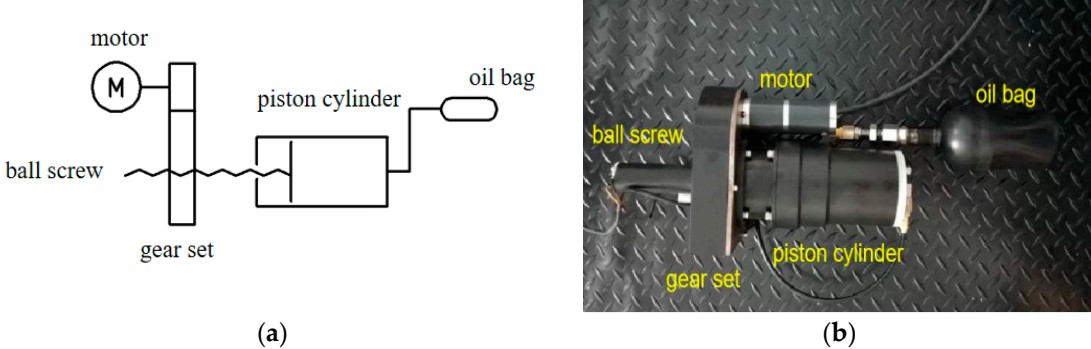

(**a**)  (**b**)

**Figure 8.** (**a**) Schematic diagram of the buoyancy adjustment system; (**b**) picture of the buoyancy adjustment system.

### 4.4. Positioning and Navigation System

The positioning and navigation system is important for the highly maneuverable AUH, because accurate positioning is necessary for accurate motion control.

The positioning system used on S-AUH is the inverse ultra-short baseline system (iUSBL). The theory and composition of iUSBL system is similar to the ultra-short baseline (USBL) system. The difference is that one end of iUSBL, which has a hydrophone array, is installed on the AUH instead of on a surface vessel that can be localized by GPS. In addition, the other end of the iUSBL system is installed on an underwater base station. iUSBL can output positioning data directly to the vehicle, while USBL should transfer data from the surface vessel to underwater vehicle through water, which is time-consuming [21].

### 4.5. Control System

The control system is used to adjust attitude and move along paths, using the positioning information received from iUSBL in real time. The control system is designed around a STM32F103VCT6 microprocessor based on an ARM-M3 kernel, which integrates the depth meter, the altimeter and the IMU (Inertial measurement unit) sensor and power output unit interface, buoyancy adjustment unit interface, positioning system interface and energy unit interface.

The control system block diagram is shown in Figure 9. The AUH needs to complete the translation and yaw steering functions in the XOY plane, the lifting and pitch control in the XOZ plane. The yaw angle is controlled by the horizontal propellers' differential output, and the pitch angle is controlled by the vertical propellers' differential output. The translation in the horizontal direction is controlled by the open loop of the horizontal propellers. The heave motion is controlled by the buoyancy adjustment system. Considering that the interference of the yaw angle and pitch angle is relatively small when

working normally, only simple coupling treatments can be carried out for yaw and pitch. When the velocity component is superimposed on the angle control component, the angle control PID (proportional-integral-derivative) parameters should be associated with the velocity component, and different angle PID parameters should be taken at different conditions. Because of the heavy workload of establishing continuous function correspondence, a piecewise PID algorithm is adopted to ensure the control accuracy.

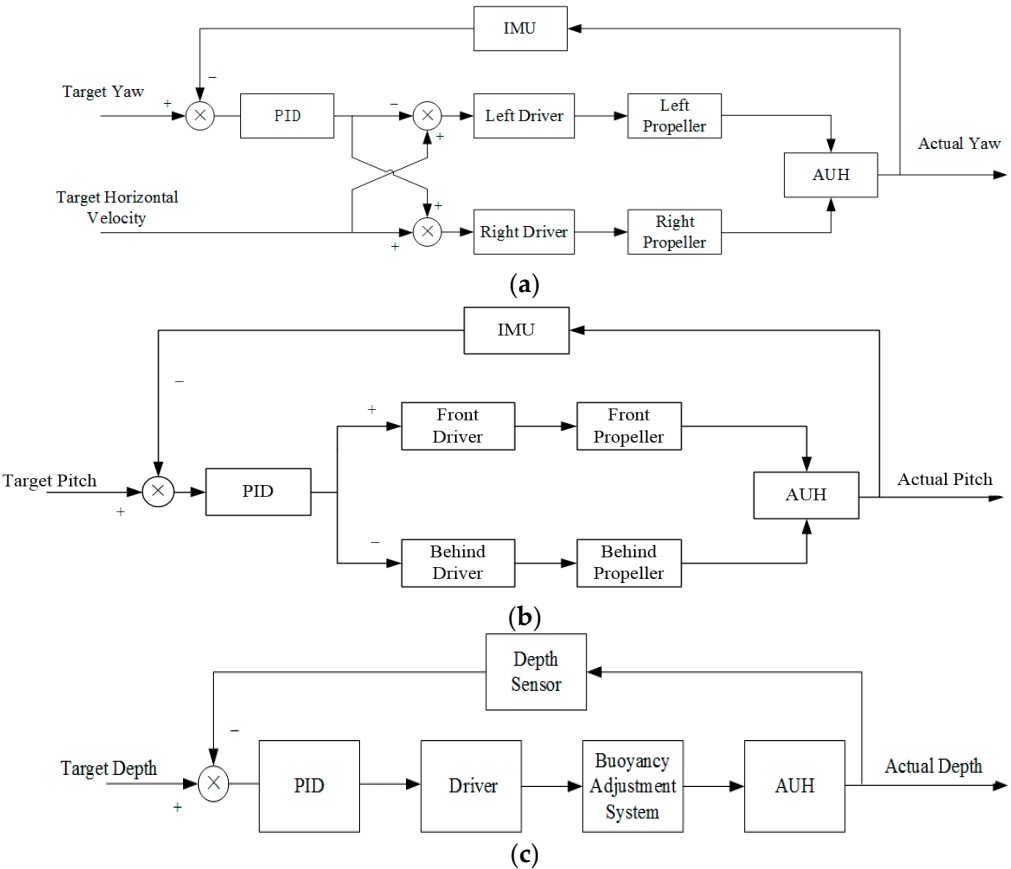

**Figure 9.** Control block diagram of (**a**) horizontal propellers; (**b**) vertical propellers; (**c**) buoyancy adjustment system.

As discussed in Section 2, it is important to have four propellers and a buoyancy adjustment system for the disc-shaped AUH to be highly maneuverable. Therefore, in this section, an AUH prototype with four propellers and a buoyancy adjustment system was developed. In addition, it will be tested in the next section to demonstrate its maneuverability.

## 5. Experiments and Results

To test the maneuverability of AUH, the horizontal linear motion experiment, horizontal rotation motion experiment and vertical lifting experiment of AUH were carried out respectively in a static pool.

In the horizontal linear motion experiment, only a pair of horizontal propellers were used to make AUH move in the horizontal plane. The experiment was carried out in a pool that was 1.4 m deep and 50 m, with slight waves. The aim of this experiment is to determine the velocity and attitude of AUH under different thrust. In four sets of experiments, we set the control code of propellers' output as 50, 100, 150, 200, respectively. The corresponding output force is 0.7 N, 1.3 N, 2.2 N, 3.2 N according to the performance of the propeller. The speed of each two recording points is calculated by the shore marking point timing, and the distance between every two recording points is 1 m. At the same time, a built-in nine-axis attitude sensor is used for attitude measurement and recording. The experimental

results are shown in Table 2. According to the results, the trend diagram of thrust and AUH velocity is obtained as shown in Figure 10. The curve form basically accords with the relationship between resistance and speed, and it can be seen that the maximum thrust 38 N propeller is enough for AUH to reach the designed speed of 0.8 m/s. It is easy to see from the table that no matter what the thrust is, AUH can basically enter the target speed within 1 m and maintain a uniform motion, which indicates that AUH has high linear motion maneuverability.

**Table 2.** Results of horizontal linear motion experiment.

| Point | | 1 | 2 | 3 | 4 | 5 | 6 | 7 | 8 | 9 | |
|---|---|---|---|---|---|---|---|---|---|---|---|
| Fx | No. | | | | Time between Two Points (s) | | | | | | Average Velocity (m/s) |
| 0.7 N | 1 | 16.0 | 11.8 | 10.2 | 10.0 | 9.9 | 9.8 | 10.3 | 11.6 | 10.7 | 0.09 |
| | 2 | 17.5 | 11.7 | 10.6 | 9.5 | 10.3 | 9.5 | 9.3 | 10.3 | 10.4 | 0.10 |
| | 3 | 20.4 | 12.3 | 10.5 | 9.0 | 8.3 | 9.3 | 8.4 | 9.3 | 10.4 | 0.10 |
| 1.3 N | 4 | 13.5 | 8.2 | 7.6 | 7.2 | 7.9 | 6.7 | 7.0 | 7.2 | 6.3 | 0.14 |
| | 5 | 15.8 | 10.2 | 7.9 | 8.1 | 7.6 | 7.1 | 7.0 | 6.3 | 6.0 | 0.13 |
| | 6 | 14.6 | 9.2 | 7.3 | 7.2 | 6.5 | 7.0 | 6.3 | 5.9 | 6.4 | 0.14 |
| 2.2 N | 7 | 7.1 | 5.5 | 4.4 | 4.6 | 3.9 | 4.3 | 4.7 | 4.4 | 4.1 | 0.22 |
| | 8 | 8.4 | 5.6 | 4.3 | 4.1 | 5.2 | 4.2 | 4.3 | 4.3 | 4.1 | 0.22 |
| | 9 | 8.0 | 5.4 | 4.8 | 4.6 | 4.2 | 4.9 | 4.4 | 4.7 | 4.3 | 0.21 |
| 3.2 N | 10 | 6.7 | 4.3 | 4.2 | 3.2 | 3.8 | 3.5 | /[1] | / | / | 0.26 |
| | 11 | 6.2 | 4.2 | 3.5 | 3.3 | 3.3 | 3.2 | / | / | / | 0.29 |
| | 12 | 6.9 | 4.0 | 3.6 | 3.4 | 4.0 | 3.7 | / | / | / | 0.27 |

[1] The experiments were terminated, because AUH touched the bottom of the pool owing to the pitching vibration.

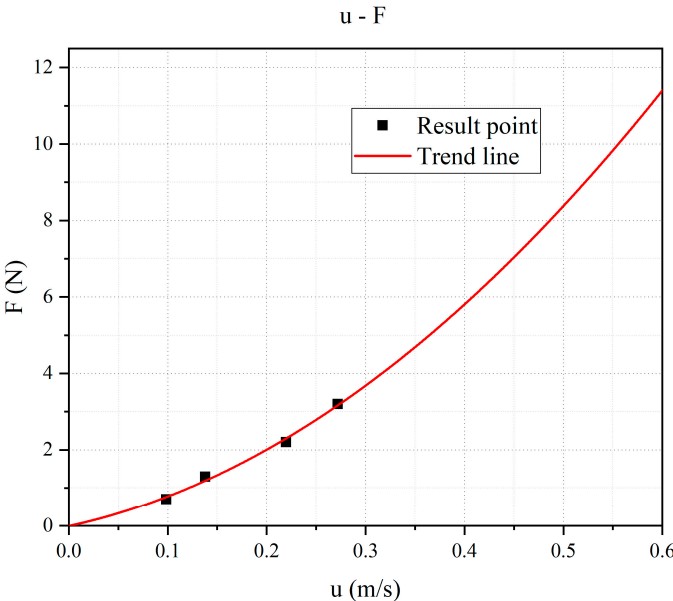

**Figure 10.** The trend diagram of thrust force F and velocity u.

The horizontal motion attitude recorded by the attitude sensor is shown in Figure 11. When only the horizontal propellers were turned on, AUH appeared pitch vibration. In addition, with the increase of thrust, the frequency and amplitude of vibration also increased. Although the of F = 2.2 N and F = 3.2 N groups had finished running before they were stable, it could be seen from the F = 1.3 N group that the pitch oscillation was a damped vibration, and that it eventually maintained a fixed non-zero pitch angle. This is consistent with the theoretical analysis in Section 2, in which vertical propellers are important for maintaining high maneuverability.

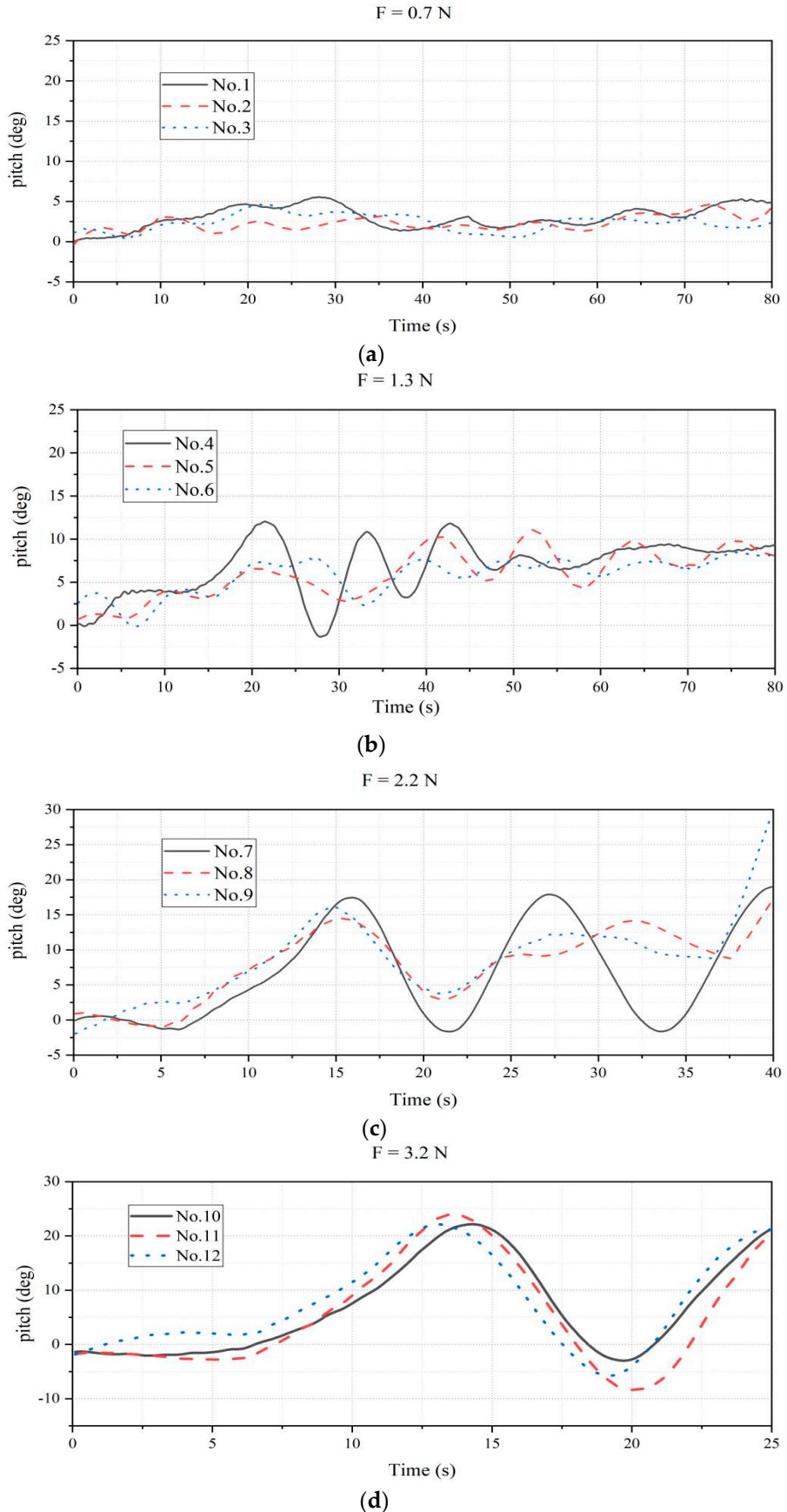

**Figure 11.** The pitch angle of horizontal motion experiments when F is (**a**) 0.7 N; (**b**) 1.3 N; (**c**) 2.2 N; (**d**) 3.2 N.

To test the effect of the vertical thruster on smooth horizontal motion, the third group of working conditions with the greatest change in pitch angle is selected as the basic working condition, that is, the thrust of the horizontal thruster is maintained at 3.2 N. The results are shown in Figure 12. Compared with the case without the vertical thruster, the vibration amplitude of the pitch angle decreases significantly. It can be seen that the vertical thruster has an obvious effect on the smooth horizontal motion.

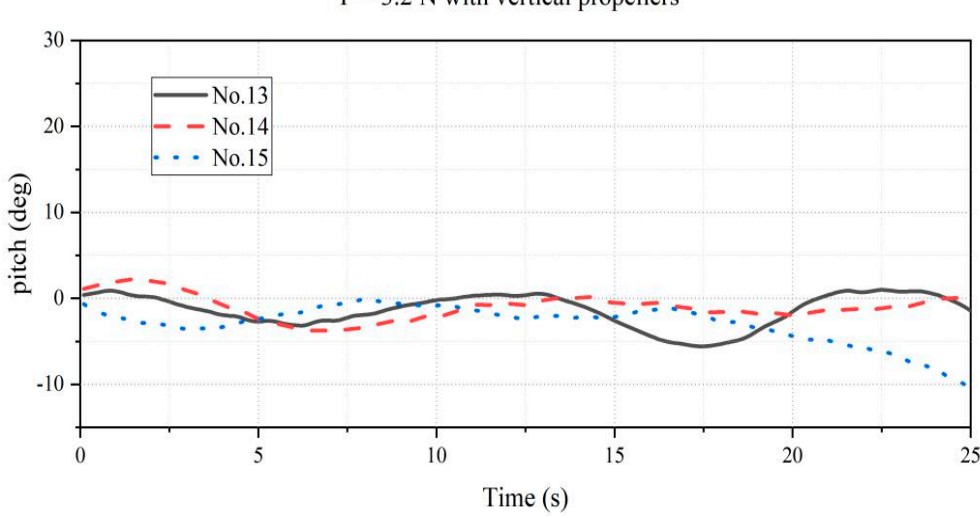

**Figure 12.** The pitch angle of horizontal motion experiments when F is 3.2 N with vertical propellers.

The purpose of the horizontal rotation experiment was to verify the zero radius rotation ability and explore the rotation response of the AUH, so as to demonstrate its maneuverability. The horizontal rotation experiment used the AUH to maintain a fixed heading direction. The AUH automatically rotated an angle and maintained it. The variation of the yaw angle during rotation was recorded by an attitude sensor. The step response is shown in Figure 13, and the average rotation speed is about 20 deg/s, almost four times as much as BOOMERANG. According to the observation, AUH had no obvious displacement when rotating. Therefore, we recognized that it does have the maneuverability of zero radius rotation.

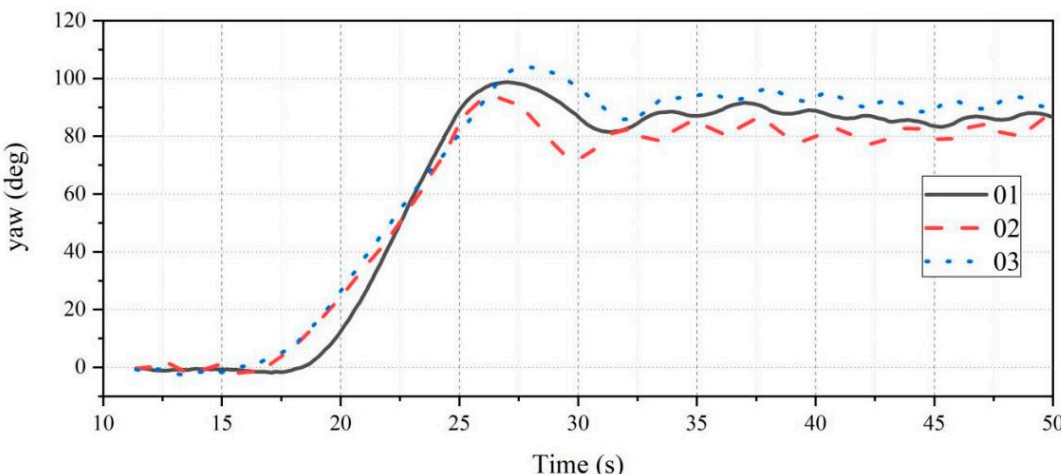

**Figure 13.** The step response of the yaw angle in the rotation experiment.

To demonstrate maneuverability, a quadrangle motion experiment in the horizontal plane was carried out. In the experiment, the AUH was controlled by the program along a rectangular path. As shown in Figure 14, the moving target of rectangular trajectory had been basically completed in the order of (a), (b), (c), (d). The AUH could cruise along a particular curve in a small area. Therefore, the maneuverability of the plane is confirmed.

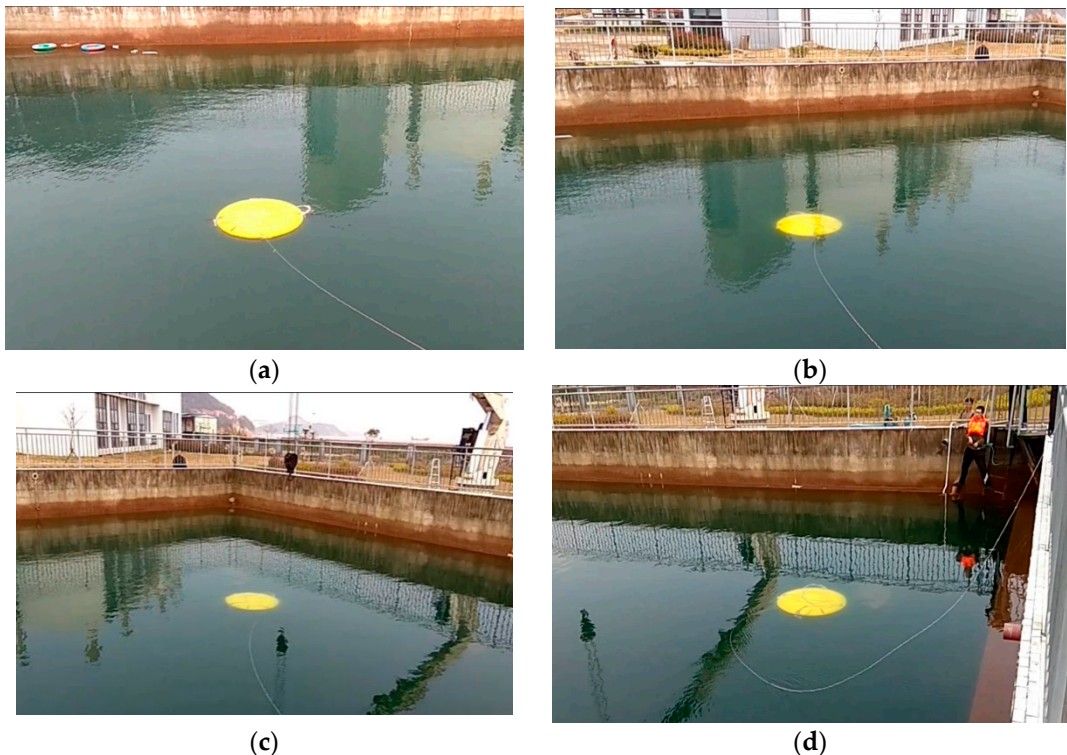

**Figure 14.** Photo of the quadrangle motion experiment in the horizontal plane: (**a**) the initial state of the quadrangle motion; (**b**) the first turn of quadrilateral motion; (**c**) the second turn of quadrilateral motion; (**d**) the third turn of quadrilateral motion.

To further demonstrate its maneuverability, an attitude sensor was used to record the attitude angle and the triaxial acceleration of the AUH, so as to estimate and evaluate the motion under the preset path experiment. The recorded attitude angle data could be obtained as shown in Figure 15a after low-pass filtering. The roll and pitch angle were relatively stable. The roll angle was small, generally below 3 degree, with a maximum of 5 degrees. The pitch angle was also controlled at around 10 degree. The trend of the yaw angle was consistent with the preset path, that is, 90 degrees per 60 s. Combined with the recorded three-axis acceleration data, an estimate of the AUH motion track was shown in Figure 15b. In the absence of position correction, AUH could still basically complete the preset path, and it was easy to see that it had excellent steering characteristics, which proves that AUH has the potential for high maneuvering performance in the horizontal plane.

The vertical lifting motion experiment was used to demonstrate the landing ability of AUH. As shown in Figure 16, the buoyancy adjusting device was used to adjust the buoyancy and complete the landing process. The AUH roll angle and pitch angle were recorded using the attitude sensor, and the results are shown in Figure 16e. Throughout the whole process, the AUH's roll angle and pitch angle were very stable. Except for the impact caused by landing at about 400 s, they were both maintained near 0 degrees, with a maximum of 2 degrees. The overall landing process was stable, showing the stability of the AUH in the vertical direction.

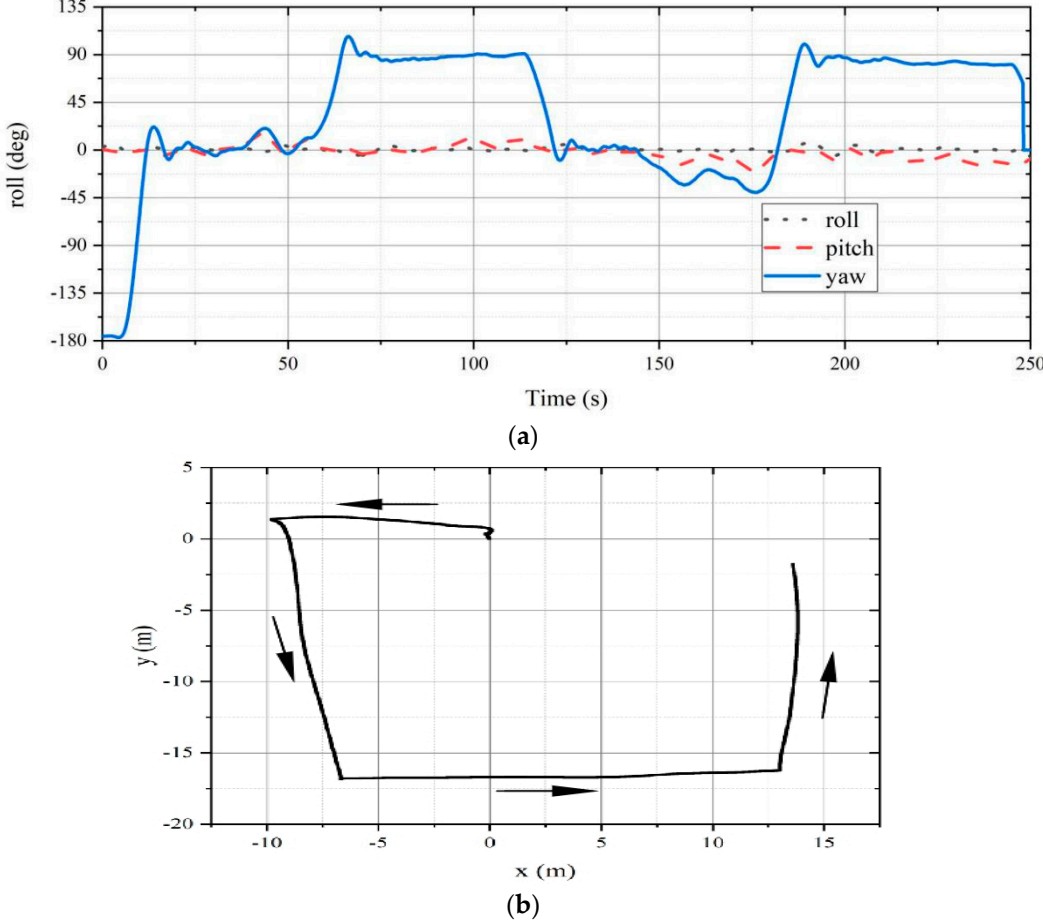

**Figure 15.** The quadrangle motion experiment in the horizontal plane. (**a**) The pitch, roll and yaw angle of AUH during the quadrangle motion experiment; (**b**) estimation of the trajectory of AUH.

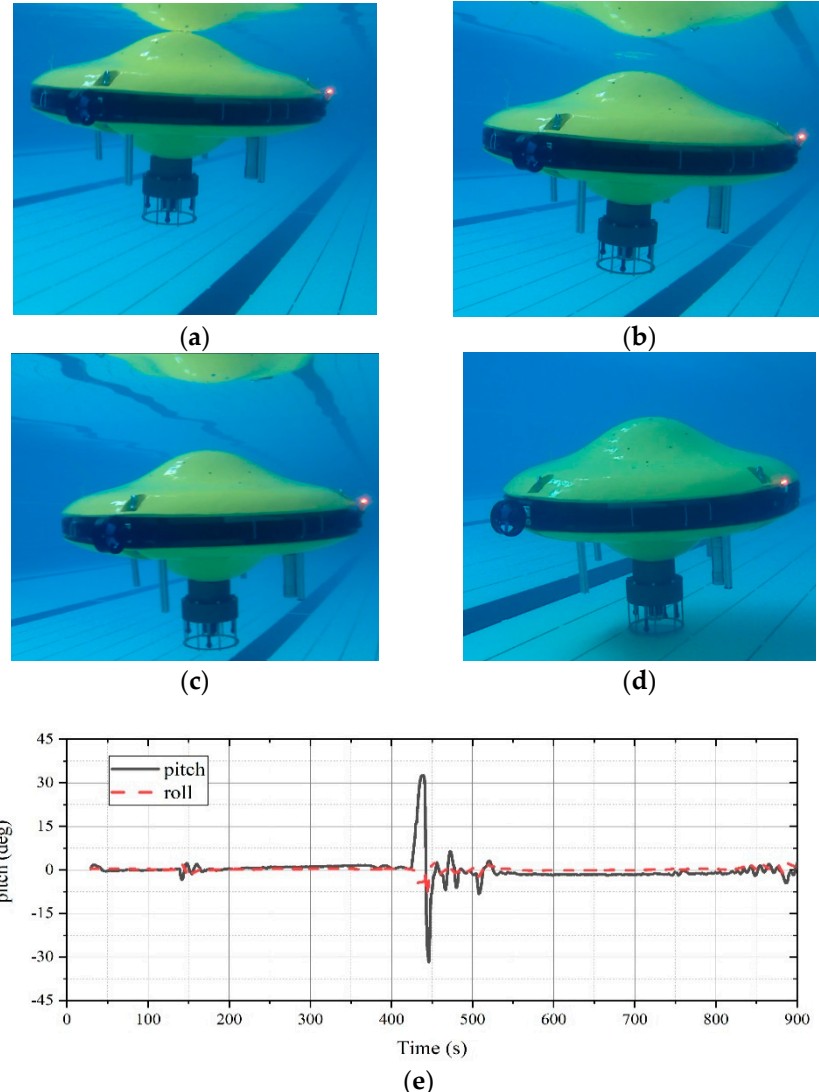

**Figure 16.** The vertical lifting motion and landing experiment. (**a**) AUH was near the surface of the water; (**b**) AUH began to sink; (**c**) AUH continued to sink; (**d**) AUH landed at the bottom of the pool; (**e**) the pitch angle and roll angle of AUH during the experiment.

## 6. Conclusions

In this paper, the AUH, a new type of disc-shaped AUV for seabed-to-seabed operating modes, is proposed. It has the characteristics of maneuverability and vertical motion stability. The prototype was built, and several groups of experiments were carried out on it. Based on this, we draw the following conclusions:

(1)    Because the special disc shape has the characteristics of horizontal in-plane isotropy, large vertical resistance, and small horizontal resistance, there is no need for over-actuation to achieve stable motion with high maneuverability within acceptable limits. It is feasible to increase endurance by reducing the number of propellers.

(2)    At least four propellers are necessary for maneuverability of the AUH. The pitch vibration in the horizontal linear motion experiment is consistent with the prediction of dynamic analysis. To eliminate the harmful resistance and angle of attack caused by this vibration, a set of propellers that can provide an active recovery moment is required. Together with at least two propellers needed to complete the plane motion, four propellers in two groups are necessary for AUH to complete the stable motion of high maneuverability.

(3)   The prototype of the AUH has the potential for high maneuverability. It has the potential to cruise in a small area flexibly, land on the seabed and take off. It is indicated that the AUH may be a solution for increasingly complex undersea engineering tasks, especially near-seabed operations.

At present, although the AUH has the potential for high maneuverability, only a simple quadrangle path test has been carried out, due to the lack of a high-accuracy positioning system and navigation system. Next, we will try to set up a more complex running path for the AUH in conjunction with the positioning and navigation system to further demonstrate its excellent maneuverability. In addition, with the high-accuracy positioning system, we will be able to more intuitively compare the performance of other submersibles, including torpedo-type submersibles, to further verify the advantages of the high maneuverability of the disc-shaped AUH. In addition, it is expected to carry working modules such as manipulators and to break through the problem that AUVs cannot carry out near-seabed engineering operations.

**Author Contributions:** The work described in this article is the collaborative development of all authors. Conceptualization, Y.C. and H.H.; methodology, Z.W.; investigation, Y.C. and H.H.; experiments performance, Z.W. and X.L.; data curation, X.L.; supervision, H.H.; writing—original draft preparation, Z.W.; writing—review and editing, H.H. and Y.C.

**Funding:** The study was supported financially by the National Key R&D Program of China (NO. 2017YFC0306100).

**Acknowledgments:** The authors would like to thank Wu Ji, Yu Zhou, and other staff of Ocean College, Zhejiang University for making the prototype and performing the experiments.

**Conflicts of Interest:** The authors declare no conflict of interest.

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
