# Peer review of "Development of an Autonomous Underwater Helicopter with High Maneuverability"

_applsci, doi:10.3390/app9194072_

Round 1
Reviewer 1 Report
Comments included in PDF. Extensive editing required for grammar and English-language usage.

Author Response
Thanks a lot for your thoughtful suggestions and insights. The manuscript has benefited from these insightful suggestions. The main comments and answers as follows:
1.The word and grammar modification suggestions
Answer: Thank you very much for such detailed modification suggestion. This error has been corrected in revised version.
The terms "large-scale" and "small-scale" need to be explained. Else, different terms need to be used to describe them.
Answer: Thank you very much for suggestion. In revised version, "several kilometers" and "a few meters" are used instead of "large-scale" and "small-scale" to make the expression more accurate.
What does "glider" form mean in this context? This has to be explained.
Answer: Thank you very much for suggestion. The glider form of AUV is that use only buoyancy and gravity to long-distance navigation similar to the principle of a glider in the air. And it has been added in revised version.
"Driving system" - needs to be replaced either with "propulsion system" or "drive system"
Answer: Thanks very much for the correction. This error has been corrected in revised version.
p, q, and r are standard symbols for angular rates (roll, pitch, and yaw rates) about body-fixed axes. They should not be used to denote the x, y, and z axes of the inertial frame.
Answer: Thanks very much for the correction. This error has been corrected in revised version.
If "u" is the velocity along x-axis, then the derivative should be the first derivative du/dt. Also, it should be a proper derivative, not a partial derivative
Answer: Thanks very much for the correction. This error has been corrected in revised version.
Authors need to further explain the claim that resistance to rotational motion will be zero. What about damping moments due to rotation rates due to the viscosity of water? To claim that this is negligible, the authors need to show a sample calculation, e.g., showing that the moment generated due to viscosity due to rotation at the design rotation rate is negligible in comparison to the thruster moments.
Answer: Thanks very much for suggestion. The resistance of AUH is mainly inertial fluid force and viscous fluid force. Since inertial fluid force can be regarded as additional mass, we mainly discuss viscous fluid force. Viscous fluid force can be divided into viscous resistance related to the velocity and contact area of the submarine and differential pressure resistance related to the shape of the submarine. Because the AUH presents a remarkable non-streamlined line, this may make the viscous resistance in translation direction negligible for differential pressure resistance. We get Equation (6). In the rotary motion, due to the rotatory characteristics of the AUH, there is only viscous resistance and no differential pressure resistance. So we can get Equation (7). Since and are only related to fluid properties and shell material, they are of the same order of magnitude. When is the same as , is the same order of magnitude as . The AUH design translation speed is 0.8m/s. Assuming that the AUH edge has a rotational linear velocity of 0.8m/s, its angular velocity is about 90 deg/s, which is sufficient for high maneuverability. The thrust of the propeller selected according to the translational design speed is at least , and the minimum torque required to maintain the design rotational speed is much smaller than the torque that the propeller can provide. Therefore, can be ignored in the analysis.
The discussion above is added in Section 3.2.
In section 3.2, it needs to be weight for the equation to be dimensionally consistent. If G is acceleration due to gravity, then it needs to be multiplied by mass to give weight.
Answer: Thanks very much for suggestion. G is the gravity force in Eq (6) not the acceleration.
A few steps need to be shown to establish that the motion is indeed a damped one. This can be shown by casting the moment equation above in a standard form and finding eigenvalues. This must be shown in the revision.
Answer: Thanks very much for suggestion. In order to more clearly illustrate the pitch motion, Equation (9) (10) is obtained by bringing Equation (8) into Equation (7), and the differential equation of motion for the pitch motion is obtained to get Equation (11). Equation (11) is sufficient to show that the pitch motion is an oscillating motion. This discussion is added in section 3.2 in the revision.
See comment regarding dynamic stability in the section on pitching motion.
Answer: Thanks very much for suggestion. Similar to pitch motion, the correction has been taken in revised version.
Reviewer 2 Report
There is a need to incorporate a few more literatures pertaining to underwater gliders :
Singh, Yogang, S. K. Bhattacharyya, and V. G. Idichandy. "CFD approach to modelling, hydrodynamic analysis and motion characteristics of a laboratory underwater glider with experimental results." Journal of Ocean Engineering and Science 2.2 (2017): 90-119.
Bustos, C., et al. "CFD Modeling of the Hydrodynamics of the CIDESI Underwater Glider." OCEANS 2018 MTS/IEEE Charleston. IEEE, 2018.
Why are hydrodynamic parameters not being used in a control system design?
I can see a cable lying along with the vehicle in experimental pictures. Is the vehicle also controlled through cables? If yes, then has the drag from the cable is being considered?
Why is experimental pitch and yaw against time are plotted? Where is the reference values from the mathematical model stated earlier in the paper ? Include and compare the simulation with experiment
Why is the trajectory of the vehicle not plotted? Include that as well.
Author Response
We thank you very much for your thoughtful suggestions and insights. The manuscript has benefited from these insightful suggestions.
There is a need to incorporate a few more literatures pertaining to underwater gliders :Singh, Yogang, S. K. Bhattacharyya, and V. G. Idichandy. "CFD approach to modelling, hydrodynamic analysis and motion characteristics of a laboratory underwater glider with experimental results." Journal of Ocean Engineering and Science 2.2 (2017): 90-119.
Bustos, C., et al. "CFD Modeling of the Hydrodynamics of the CIDESI Underwater Glider." OCEANS 2018 MTS/IEEE Charleston. IEEE, 2018.
Answer: Thanks very much for suggestion. The references have been added in reference section in revised version.
Why are hydrodynamic parameters not being used in a control system design?
Answer: Because the control system uses a modeless PID control method. Hydrodynamic parameters are only used for shape design using theoretical analysis and simulation analysis, but not used in the establishment of control models.
I can see a cable lying along with the vehicle in experimental pictures. Is the vehicle also controlled through cables? If yes, then has the drag from the cable is being considered?
Answer: The cable visible in experimental picture is only a safety rope and will not be used in practical applications.
Why is experimental pitch and yaw against time are plotted? Where is the reference values from the mathematical model stated earlier in the paper? Include and compare the simulation with experiment
Answer: The plot of the pitch angle against time is used to verify the prediction that the horizontal advancement of AUH in the theoretical analysis may produce a prediction of the pitch direction oscillation. And demonstrate the necessity of the vertical thruster to maintain a smooth progression of the AUH. The Yaw against time is used to illustrate the maneuverability of the AUH steering, demonstrating its superior steering performance in similar AUVs.
The theoretical reference is in the newly added equations (5) and (11) added in Section 3 and the subsequent interpretation analysis.
Since many important parameters such as weight, centroid distribution, and floating center distribution have made great changes in the early stage simulation construction, the simulation results have no numerical reference value, so they are not described in this paper.
Why is the trajectory of the vehicle not plotted? Include that as well.
Answer: Thanks very much for suggestion. Lack of corresponding observing equipment. Because it is an indoor experiment, it is impossible to use a drone to obtain a panoramic image, and it is impossible to draw an accurate trajectory. The navigation system cannot be used in shallow experimental pools, so AUH does not have accurate positioning movements. It is of little significance to draw the trajectory. Therefore, only the video recording the approximate path is left. Accurate specified path experiments and external reference path maps will be further confirmed in the next possible sea trial.
Reviewer 3 Report
The current study proposes a disc-shaped underwater vehicle intended for executing maneuvers near the ocean floor. Although the study presents an interesting vehicle design that may be advantageous over competing designs, various issues hindering the acceptance of the manuscript are stated in the comments below.
Comment 1: Figure 1 shows a device labeled "ZJU Subsea Station," although this device is not directly mentioned in the text. The purpose of this station would be clearer to the reader, if the station is explained in the caption of the figure or in the main text.
Comment 2: Figure 1 contains curves or arrows that are assigned to each vehicle. The meaning of the curves and arrows should be clearly indicated in the caption or in the figure, since it is not clear if the curves indicate work patterns or something else.
Comment 3: The claim is made in lines 84 and 85 that the requirements of the main working pattern of the AUH are met only if the AUH is "maneuverable in the horizontal plane and stable in the vertical direction." Justification for this claim should be provided in Chapter 2 to improve the logical validity of the manuscript.
Comment 4: On lines 91 to 92, the word "destroy" seems rather strong. Consider changing the word to "compromise."
Comment 5: Unlike Equations 1 and 2, the vector variables in Equations 4 through 9, such as u and Jz, are not written in bold to indicate that they are vectors. For the sake of consistency across all equations, it is recommended to express any vector variables in bold.
Comment 6: In it not clear whether Figure 6a is a top or bottom view of the AUH.
Comment 7: Figure 6b should be labeled, as in the case of 6a, to identify internal AUH parts.
Comment 8: The phrase "track tracking" seems awkward, since underwater vehicles move along paths instead of tracks.
Comment 9: According to Figure 9a, if L = 1000 mm, then the values for l1, l2, and l3 should be 125 mm, 250 mm, 375 mm, respectively.
Comment 10: Line 210 mentions Figure 10, but Figure 10 is not contained in the manuscript.
Comment 11: Figure 9 comes before Figure 7 in the manuscript.
Comment 12: Major components of the buoyancy adjustment system should be labeled in Figure 7b, as in the case of Figure 7a.
Comment 13: Although a publication is cited in Line 220 to support the selection of the propeller in Table 1, it would be more convincing to provide the reason for the selection of the propeller in the main body of the manuscript.
Comment 14: Lines 202 and 203 state the "design speed" and "design depth," although no reason is provided for the selection of these parameters. Furthermore, Figure 1 shows the AUH at different depths and Equation 4 indicates that different speeds are possible. Therefore, it is not obvious how the design depth and design speed describe the normal operation of the vehicle. It would be more descriptive to also provide the range of speed and the range of depth at which the AUH operates.
Comment 15: Unabbreviated names for IMU and PID are not provided. Since PIDs and IMUs are technical terms, it is important to provide the complete names of the devices as a courtesy to readers who are not familiar with these terms.
Comment 16: Figure 11 indicates the speed of the AUH in m/s, even though the design speed is stated in knots. In order to make interpretation of the results easier, it is recommended to state the speed only in knots or in m/s.
Comment 17: Figure 13 shows the step response of the yaw angle, although no standard performance parameters such as settling time and steady state error are provided. Furthermore, it is not clear from Figure 13, if the yaw angle has reached steady state.
Comment 18: The horizontal rotation experiment appears to only have been performed once, and thus no average or mean step response curve for yaw is provided. However, in order to accurately evaluate the repeatability of the horizontal rotation of the AUH, the average or median step response curve should be provided, along with a measures of variability such as standard deviation curves or variance curves.
Comment 19: The quadrangle motion experiment was performed to assess the maneuverability of the AUH in the horizontal plane. However, the text does not explain how the position of the AUH was measured, nor does the text provide any quantitative criteria for confirming maneuverability.
Comment 20: If the working pattern of the AUH is to take off and land on the ocean floor, as shown in Figure 2, it is not clear how the quadrangle motion experiment, which is performed near the surface of a pool instead of the ocean floor, is relevant to the working pattern.
Comment 21: Although the landing process was observed to be stable in a swimming pool during the vertical lifting motion experiment, the terrain of ocean floors can vary widely in stability. Since the working pattern of the AUH is performed near the ocean floor, the vertical lifting motion experiment would be more convincing and accurate if performed in the ocean or in a pool that simulates the terrain of the ocean floor.
Comment 22: The first conclusion on Line 320 claims that it is "feasible to increase endurance by reducing the number of propellers." None of the results directly supports this claim because reducing the number of propellers was not performed for any of the experiments.
Comment 23: Figure 12 shows the pitch of the AUH when only the horizontal propellers are operating. However, the pitch when all of the propellers are turned on is not shown. Although it is claimed that the vertical propellers are important for maintaining high maneuverability, no data is provided to directly demonstrate the effect of all propellers on the pitch of the vehicle.
Comment 24: The third conclusion on Lines 328 to 330 claims that the AUH can land on the seabed and take off, even though the landing process for the vertical lifting motion experiment was performed on the floor of a pool rather than an actual seabed.
Comment 25: Dividing lines added to Table 2 would clearly distinguish the groups of experimental trials that belong to each propeller output setting.
Comment 26: The design depth is specified as 100 m, although no depth is provided for the pool in the vertical lifting motion experiment, nor does the depth of this pool appear to be 100 m.
Comment 27: Aside comparing the rotation speed of the BOOMERANE to the proposed AUH, no other detailed quantitative comparisons are made between the AUH and other underwater vehicles. The extent to which the AUH has greater maneuverability than other vehicles is therefore unclear.
Concluding remarks: The experiments were performed in an a pool environment, rather than the actual ocean, and thus provide a preliminary evaluation of the maneuverability of the AUH. The methodological justification for these experiments is lacking, since it is not explained how vertical stability and horizontal maneuverability are necessary to realize the work pattern of the AUH in the actual ocean. Furthermore, it is not explained how a pool environment sufficiently simulates ocean conditions under which the work pattern is performed.
Since more trials are needed to more accurately assess the horizontal rotation of the AUH, and no quantitative data is provided for the quadrangle motion experiment, more data is necessary to evaluate the horizontal maneuverability of the AUH. Furthermore, it is claimed on Line 320 that endurance can be increased by reducing the number of propellers, although experimental data is needed to support this claim.
Consequently, the results do not sufficiently support the conclusions of the study. Based on this insufficiency and numerous issues relating to argument structure, data presentation, figure formatting, spelling, grammar, and word usage, I have regrettably decided not to accept the current manuscript. It is recommended that more experiments be performed and that the manuscript be revised and proofread extensively to be considered for publication at another time.
Author Response
We thank you very much for your thoughtful suggestions and insights. The manuscript has benefited from these insightful suggestions.
Figure 1 shows a device labeled "ZJU Subsea Station," although this device is not directly mentioned in the text. The purpose of this station would be clearer to the reader, if the station is explained in the caption of the figure or in the main text.
Answer: Thanks very much for suggestion. ZJU Subsea Station is the docking base station for AUH for data transmission and power transmission. Instructions have been added to the description under the image in revised version.
Figure 1 contains curves or arrows that are assigned to each vehicle. The meaning of the curves and arrows should be clearly indicated in the caption or in the figure, since it is not clear if the curves indicate work patterns or something else.
Answer: Thanks very much for suggestion. The blue line is working paths of AUV, float, and Gilder. Yellow line is the working path of AUH. Red line is AUH’s expected work sequence. The above instructions have been added to the figure.
The claim is made in lines 84 and 85 that the requirements of the main working pattern of the AUH are met only if the AUH is "maneuverable in the horizontal plane and stable in the vertical direction." Justification for this claim should be provided in Chapter 2 to improve the logical validity of the manuscript..
Answer: Thanks very much for suggestion. The focus of this paper is on the maneuverability of the dish-shaped AUH, and the vertical stability is not the focus of the article. Therefore, in the analysis of the vertical motion in Section 3.3, the vertical stability is briefly analyzed.
The claim is made in lines 84 and 85 that the requirements of the main working pattern of the AUH are met only if the AUH is "maneuverable in the horizontal plane and stable in the vertical direction." Justification for this claim should be provided in Chapter 2 to improve the logical validity of the manuscript.
Answer: Thanks very much for suggestion. Accurate motion above the base station requires AUH to have good maneuverability in the horizontal plane, while stable hovering over the base station requires AUH stability in the vertical direction. It has been added in revised version.
Unlike Equations 1 and 2, the vector variables in Equations 4 through 9, such as u and Jz, are not written in bold to indicate that they are vectors. For the sake of consistency across all equations, it is recommended to express any vector variables in bold.
Answer: Thanks very much for suggestion. Equations (4)-(13) are equations in one direction in the coordinate system. All variables in the equation are scalars of the components of the vector in that direction. This succinct and clear scalar form is used because the analysis is performed on certain axes or planes.
In it not clear whether Figure 6a is a top or bottom view of the AUH.
Answer: Thanks very much for suggestion. Figure 6a is a top view of AUH. This error has been corrected in revised version.
Figure 6b should be labeled, as in the case of 6a, to identify internal AUH parts.
Answer: Thanks very much for suggestion. Figure 6b has been labeled in revised version.
The phrase "track tracking" seems awkward, since underwater vehicles move along paths instead of tracks.
Answer: Thanks very much for correction. This error has been corrected in revised version.
According to Figure 9a, if L = 1000 mm, then the values for l1, l2, and l3 should be 125 mm, 250 mm, 375 mm, respectively.
Answer: Thanks very much for correction. This error has been corrected in revised version.
Line 210 mentions Figure 10, but Figure 10 is not contained in the manuscript.
Answer: Thanks very much for correction. This error has been corrected in revised version.
Figure 9 comes before Figure 7 in the manuscript.
Answer: Thanks very much for correction. This error has been corrected in revised version.
Major components of the buoyancy adjustment system should be labeled in Figure 7b, as in the case of Figure 7a..
Answer: Thanks very much for suggestion. The components in Figure 7b has been labeled in revised version.
Although a publication is cited in Line 220 to support the selection of the propeller in Table 1, it would be more convincing to provide the reason for the selection of the propeller in the main body of the manuscript.
Answer: Thanks very much for suggestion. According to the results of hydrodynamic simulation of AUH in [20], the T200 propeller with performance as shown in Table 1 is selected. Its output thrust range meets the requirements for use because by simulation analysis[20], it is considered that the thrust required for the design speed of 1 section is 15N. And the propeller we chose is light in weight, small in size and simple in mechanical connection. The reason above has been added in revised version.
Lines 202 and 203 state the "design speed" and "design depth," although no reason is provided for the selection of these parameters. Furthermore, Figure 1 shows the AUH at different depths and Equation 4 indicates that different speeds are possible. Therefore, it is not obvious how the design depth and design speed describe the normal operation of the vehicle. It would be more descriptive to also provide the range of speed and the range of depth at which the AUH operates.
Answer: Thanks very much for suggestion. Since the experimental prototype produced this time is only a prototype for verifying the working principle of the circular dish AUV, considering the test and iteration cost, the design depth is selected as 100m, so that the design cost is low and the experimental environment is easier to find. The design speed is the estimated average flow rate in the working environment, considering that it should have a certain resistance to flow to achieve hovering. What is shown in Figure 1 is not that AUH works at different depths, just that it works close to the seabed. The above explanation is also added in revised version.
Unabbreviated names for IMU and PID are not provided. Since PIDs and IMUs are technical terms, it is important to provide the complete names of the devices as a courtesy to readers who are not familiar with these terms.
Answer: Thanks very much for suggestion. The full names of the abbreviations PID and IMU have been added in revised version.
Figure 11 indicates the speed of the AUH in m/s, even though the design speed is stated in knots. In order to make interpretation of the results easier, it is recommended to state the speed only in knots or in m/s.
Answer: Thanks very much for suggestion. The speed has been stated only in m/s in revised version.
Figure 13 shows the step response of the yaw angle, although no standard performance parameters such as settling time and steady state error are provided. Furthermore, it is not clear from Figure 13, if the yaw angle has reached steady state.
Answer: Thanks very much for suggestion. Due to cost considerations, the attitude sensor used by the prototype AUH has low accuracy and stability, and there is no ability to accurately quantify stable measurements. Therefore, no standard adjustment time and error measurements were made, only a relatively rough motion function test was performed. The second half of the fluctuations in Fig. 13 are the same as the magnitude of the disturbance and the drift of the offset and attitude sensors themselves, so it is considered that the AUH has reached a steady state at this time.
The horizontal rotation experiment appears to only have been performed once, and thus no average or mean step response curve for yaw is provided. However, in order to accurately evaluate the repeatability of the horizontal rotation of the AUH, the average or median step response curve should be provided, along with a measures of variability such as standard deviation curves or variance curves.
Answer: As described above, the accuracy and stability of the currently used attitude measurement sensor are low, and it isn’t worth of reference of the statistical characteristics obtained by repeated measurement. In further research, in a suitable acoustic experimental environment, combined with the professional iUSBL positioning navigation system, accurate tests can be performed and corresponding statistical characteristics can be obtained.
The quadrangle motion experiment was performed to assess the maneuverability of the AUH in the horizontal plane. However, the text does not explain how the position of the AUH was measured, nor does the text provide any quantitative criteria for confirming maneuverability.
Answer: Thanks very much for suggestion. The horizontal motion experiment in this paper aims to qualitatively determine that the prototype AUH can complete the quadrilateral motion including right angles to verify its mobility. Due to the limitations of the experimental environment, position measuring devices that can accurately measure the distance are not used. We will also add positioning navigation systems in the appropriate acoustic experimental environment in the future to complete accurate position determination.
If the working pattern of the AUH is to take off and land on the ocean floor, as shown in Figure 2, it is not clear how the quadrangle motion experiment, which is performed near the surface of a pool instead of the ocean floor, is relevant to the working pattern.
Answer: The AUH mode of operation consists of three phases, namely, taking off from the base station, sailing near the seabed along a predetermined path within a fixed range, and landing on the base station after reaching the next base station. The takeoff and landing functions in this process were verified in the vertical takeoff and landing experiments. The quadrangle motion experiment which is performed near the surface of a pool is to verify that it can perform more complicated actions in the near-submarine plane after take-off to reach the next base station, and can flexibly land on it. The reason why the plane motion experiment is used to verify this is because the design working environment is a relatively flat seabed area and a depth profile at a certain distance from the seabed, which can be regarded as a similar environment to the pool near the surface.
Although the landing process was observed to be stable in a swimming pool during the vertical lifting motion experiment, the terrain of ocean floors can vary widely in stability. Since the working pattern of the AUH is performed near the ocean floor, the vertical lifting motion experiment would be more convincing and accurate if performed in the ocean or in a pool that simulates the terrain of the ocean floor.
Answer: As shown in Figure 1, the working mode of AUH is to land on the base station laid on the seabed.The base station can adjust its flatness to a certain extent, making it equivalent to a plane on the seabed. Therefore, it is not necessary to consider the conditions under which AUH landed on the seabed in a complex environment.
The first conclusion on Line 320 claims that it is "feasible to increase endurance by reducing the number of propellers." None of the results directly supports this claim because reducing the number of propellers was not performed for any of the experiments.
Answer: The so-called reducing the number of propellers here means that the 4 propellers mode is an attempt to reduce the number of propellers compared to the 6~8 propellers used in ROV and other submersibles to meet the maneuverability in the conventional practice. All the experiments in this paper are also based on the AUH of 4 propellers, which can achieve a certain degree of maneuverability, indicating that it is feasible to reduce the number of propellers to 4 compared to the conventional practice. The reduction of the propellers means that the total weight and power consumption are reduced, which can increase the endurance.
Figure 12 shows the pitch of the AUH when only the horizontal propellers are operating. However, the pitch when all of the propellers are turned on is not shown. Although it is claimed that the vertical propellers are important for maintaining high maneuverability, no data is provided to directly demonstrate the effect of all propellers on the pitch of the vehicle.
Answer: Thanks very much for suggestion. A set of data from experiments using vertical thrusters was added after Figure 12 to further illustrate that vertical thrusters are indeed effective.
The third conclusion on Lines 328 to 330 claims that the AUH can land on the seabed and take off, even though the landing process for the vertical lifting motion experiment was performed on the floor of a pool rather than an actual seabed.
Answer: Thanks very much for suggestion. The AUH prototype used in this paper is intended to be verified by experiments for principle verification, and the conclusions of the article are not rigorous. The conclusion was revised to demonstrate that AUH has the potential to complete small-scale maneuvers and take off and land on submarine subsea stations.
Dividing lines added to Table 2 would clearly distinguish the groups of experimental trials that belong to each propeller output setting.
Answer: Thanks very much for suggestion. The dividing lines has been added to Table 2 in revised version.
The design depth is specified as 100 m, although no depth is provided for the pool in the vertical lifting motion experiment, nor does the depth of this pool appear to be 100 m.
Answer: In the case where the principle verification does not consider the 100m current, the AUH is most affected by the 100m design depth is its its pressure structure design. Each AUH cabins is designed according to 100m, and all of them passed the 100m pressure test. The selection of equipment such as propellers also meets the design depth requirement of 100m. This proves that the AUH can operate normally at a design depth of 100m without performing a 100m sea trial. Related instructions have been added in Section 4.
Aside comparing the rotation speed of the BOOMERANE to the proposed AUH, no other detailed quantitative comparisons are made between the AUH and other underwater vehicles. The extent to which the AUH has greater maneuverability than other vehicles is therefore unclear.
Answer: In the discussion of the Section 1 and Section 3, we mentioned that the disk-shape is high maneuverable in small-scale and stable in vertical direction, relative to the torpedo type. Therefore, in the specific indicators, it is mainly compared with the BOOMERANGE and the disc-shape glider of Dalian University of Technology, which have disc shape same as AUH. In the following work we will also compare the indicators of more types of submersibles to further verify the unique advantages of AUH in the field of near-seabed operations.
Round 2
Reviewer 1 Report
Note: Eq. 4 and Eq. 5 still show partial derivatives. These should be regular derivatives du/dt and dw/dt.
Reviewer 2 Report
The authors have revised the manuscript and it looks fine to be published.
Author Response
Thank you very much for your suggestions of great value.
Reviewer 3 Report
Please find my previous comments below with the authors' answers and my replies to these answers.
Comment 13: Although a publication is cited in Line 220 to support the selection of the propeller in Table 1, it would be more convincing to provide the reason for the selection of the propeller in the main body of the manuscript.
Answer to Comment 13: Thanks very much for suggestion. According to the results of hydrodynamic simulation of AUH in [20], the T200 propeller with performance as shown in Table 1 is selected. Its output thrust range meets the requirements for use because by simulation analysis[20], it is considered that the thrust required for the design speed of 1 section is 15N. And the propeller we chose is light in weight, small in size and simple in mechanical connection. The reason above has been added in revised version.
The article referenced in [20] is used to support the decision to select the T200 propeller. However, the results of [20] do not mention that the required thrust is 15N, nor is it clear what is meant by “1 section.”
Comment 15: Unabberviated names for IMU and PID are not provided. Since PIDs and IMUs are technical terms, it is important to provide the complete names of the devices as a courtesy to readers who are not familiar with these terms.
Answer to Comment 15: Thanks very much for suggestion. The full names of the abbreviations PID and IMU have been added in revised version.
PID stands for “proportional-integral-derivative” rather than “proportion-integral-derivative.”
Comment 17: Figure 13 shows the step response of the yaw angle, although no standard performance parameters such as settling time and steady state error are provided. Furthermore, it is not clear from Figure 13, if the yaw angle has reached steady state.
Answer to Comment 17: Thanks very much for suggestion. Due to cost considerations, the attitude sensor used by the prototype AUH has low acuracy and stability, and there is no ability to accurately quantify stable measurements. Therefore, no standard adjustment time and error measurements were made, only a relatively rough motion function test was performed. The second half of the fluctuations in Fig. 13 are the same as the magnitude of the disturbance and the drift of the offset and attitude sensors themselves, so it is considered that the AUH has reached a steady state at this time.
Comment 18: The horizontal rotation experiment appears to only have been performed once, and thus no average or mean step response curve for yaw is provided. However, in order to accurately evaluate the repeatability of the horizontal rotation of the AUH, the average or mean step response curve should be provided, along with a measures of variability such as standard deviation curves or variance curves.
Answer to Comment 18: As described above, the accuracy and stability of the currently used attitude measurement sensor are low, and it isn’t worth of reference of the statistical characteristics obtained by repeated measurement. In further research, in a suitable acoustic experimental environment, combined with the professional iUSBL positioning navigation system, accurate tests can be performed and corresponding statistical characteristics can be obtained.
If the attitude sensor has low accuracy and stability, then the conclusion on Line 331 that the AUH has “high maneuverability” is not validated with statistical significance. It is curious that this limitation is not directly acknowledged in the current manuscript, given that a foundational aspect of scientific research is repeatability, as conventionally confirmed through statistical analysis. More importantly, it remains to be explained how a rough motion function test without an objective statistical analysis could support future research. It is recommended to perform the test again with more accurate and precise measurement equipment, if the condition of financial cost is favorable in the future.
Comment 19: The quadrangle motion experiment was performed to assess the maneuverability of the AUH in the horizontal plane. However, the text does not explain how the position of the AUH was measured, nor does the text provide any quantitative criteria for confirming maneuverability.
Answer to Comment 19: Thanks very much for suggestion. The horizontal motion experiment in this paper aims to qualitatively determine that the prototype AUH can complete the quadrilateral motion including right angles to verify its mobility. Due to the limitations of the experimental environment, position measuring devices that can accurately measure the distance are not used. We will also add positioning navigation systems in the appropriate acoustic experimental environment in the future to complete accurate position determination.
Although is possible to qualitatively determine quadrilateral motion, objective analysis involving statistics would provide a stronger case for the maneuverability of the AUH. As in the case of the horizontal rotation experiment, it would be more scientifically sound to redo the quadrilateral motion test, if more accurate and precise measurements become available.
Comment 27: Aside comparing the rotation speed of the BOOMERANE to the proposed AUH, no other detailed quantitative comparisons are made between the AUH and other underwater vehicles. The extent to which the AUH has greater maneuverability than other vehicles is therefore unclear.
Answer to Comment 27: In the discussion of the Section 1 and Section 3, we mentioned that the disk-shape is high maneuverable in small-scale and stable in vertical direction, relative to the torpedo type. Therefore, in the specific indicators, it is mainly compared with the BOOMERANGE and the disc-shape glider of Dalian University of Technology, which have disc shape same as AUH. In the following work we will also compare the indicators of more types of submersibles to further verify the unique advantages of AUH in the field of near-seabed operations.
A thorough quantitative comparison remains to performed, even in the case of the torpedo type vehicle. Plans to compare parameters of different types of submersibles should be included in the conclusion or in a separate section on future work and limitations.
Concluding remarks: Based on the current manuscript, it is evident that there are major improvements, such as clearer figures and tables as well as more data to make comparisons between different states of propeller operation. Aside from many minor revisions that need to be made, such as proper word expressions, spelling, and use of capitalization for figures and tables, e.g. the legend for Figure 1, the vertical axis titles for Figure 11, etc., the manuscript still requires more substantial quantitative analysis to qualify as a scientifically sound article with conclusions that are supported by statistical data. As the authors have pointed out in their response to Comment 24, the conclusions are “not rigorous.”
Although it would be premature to consider this preliminary study for publication as an Article, it would be more fitting to convert it to a conference paper or to a Short Communication for Applied Sciences.
